# Recent Advances in Photodynamic Therapy: Metal-Based Nanoparticles as Tools to Improve Cancer Therapy

**DOI:** 10.3390/pharmaceutics16070932

**Published:** 2024-07-12

**Authors:** Stefania Mariano, Elisabetta Carata, Lucio Calcagnile, Elisa Panzarini

**Affiliations:** 1Department of Mathematics and Physics, University of Salento, 73100 Lecce, Italy; stefania.mariano@unisalento.it (S.M.); lucio.calcagnile@unisalento.it (L.C.); 2Department of Biological and Environmental Sciences and Technologies, University of Salento, 73100 Lecce, Italy; elisabetta.carata@unisalento.it; 3CEDAD (CEntre of Applied Physics, DAtation and Diagnostics), Department of Mathematics and Physics “E. De Giorgi”, University of Salento, 73100 Lecce, Italy

**Keywords:** metal-based nanoparticles, photodynamic therapy, photosensitizer, toxicity, biocompatibility, targeting, cell death

## Abstract

Cancer remains a significant global health challenge, with traditional therapies like surgery, chemotherapy, and radiation often accompanied by systemic toxicity and damage to healthy tissues. Despite progress in treatment, these approaches have limitations such as non-specific targeting, systemic toxicity, and resistance development in cancer cells. In recent years, nanotechnology has emerged as a revolutionary frontier in cancer therapy, offering potential solutions to these challenges. Nanoparticles, due to their unique physical and chemical properties, can carry therapeutic payloads, navigate biological barriers, and selectively target cancer cells. Metal-based nanoparticles, in particular, offer unique properties suitable for various therapeutic applications. Recent advancements have focused on the integration of metal-based nanoparticles to enhance the efficacy and precision of photodynamic therapy. Integrating nanotechnology into cancer therapy represents a paradigm shift, enabling the development of strategies with enhanced specificity and reduced off-target effects. This review aims to provide a comprehensive understanding of the pivotal role of metal-based nanoparticles in photodynamic therapy. We explore the mechanisms, biocompatibility, and applications of metal-based nanoparticles in photodynamic therapy, highlighting the challenges and the limitations in their use, as well as the combining of metal-based nanoparticles/photodynamic therapy with other strategies as a synergistic therapeutic approach for cancer treatment.

## 1. Introduction

Cancer, a multifaceted and pervasive disease, continues to pose a significant global health challenge. Traditional cancer therapies, including surgery, chemotherapy, and radiation, have undoubtedly contributed to progress in its treatment, but they are often accompanied by limitations such as systemic toxicity and damage to healthy tissues [1]. In recent years, the field of nanotechnology has emerged as a revolutionary frontier in cancer therapy, offering a new paradigm that holds promise in overcoming some of these challenges. Cancer therapy is not merely about eradicating malignant cells; it includes a broader spectrum of goals aimed at improving patient outcomes and enhancing their quality of life. The significance lies in achieving a delicate balance between eliminating cancerous cells and preserving the normal physiological functions of the body [2]. Moreover, as cancer often manifests as a heterogeneous and dynamic disease, the need for adaptable and targeted therapeutic strategies becomes increasingly apparent.

Despite their efficacy, traditional cancer therapies exhibit inherent limitations that necessitate a paradigm shift in treatment approaches. Chemotherapy, for instance, often results in systemic toxicity due to its non-specific targeting, leading to adverse side effects that compromise patient well-being. Additionally, the development of resistance in cancer cells poses a formidable challenge in maintaining treatment effectiveness over time [3]. Radiation therapy, while potent, can damage surrounding healthy tissues [4]. Addressing these limitations is crucial for enhancing therapeutic outcomes and minimizing the overall burden on patients undergoing treatment.

Nanoparticles, with their unique physical and chemical properties, offer a promising avenue for overcoming the limitations of traditional cancer therapies. Their small size, high surface-area-to-volume ratio, and tunable surface properties allow for precise interactions at the molecular and cellular levels. Nanoparticles can be engineered to carry therapeutic payloads, navigate biological barriers, and selectively target cancer cells [5]. Metal-based nanoparticles (MBNPs), in particular, offer unique properties that make them suitable for a variety of therapeutic applications.

The integration of nanotechnology into cancer therapy represents a paradigm shift, enabling the development of innovative strategies with enhanced specificity and reduced off-target effects. Nanoparticles can be designed to deliver therapeutic agents directly to cancer cells, improving drug bioavailability and minimizing damage to healthy tissues. Moreover, the ability of nanoparticles to target tumors holds great promise, passively or actively, for achieving localized and efficient treatment [6]. In the realm of cancer therapy, photodynamic therapy (PDT) combined with nanoparticles has emerged as a cutting-edge approach, offering improved precision and effectiveness in targeting and destroying cancerous cells. The main objective of this review is to provide a comprehensive understanding of the significance played by MBNPs in cancer therapy, particularly in the context of PDT. We will deeply explore the mechanisms, biocompatibility, and clinical applications of MBNPs in the context of PDT, as well as the combination of MBNPs in PDT with other strategies as a synergistic therapeutic approach for cancer treatment.

## 2. Metal Nanoparticles and Their Role in Cancer Therapy

MBNPs are particles composed of metal atoms, typically ranging in size from 1 to 100 nanometers. These nanoparticles exhibit unique physical, chemical, and optical properties that differ significantly from their bulk counterparts, owing to their high surface-area-to-volume ratio and quantum confinement effects [7]. One of the most intriguing aspects of MBNPs is their tunable properties, which can be adjusted by controlling their size, shape, composition, and surface chemistry [8,9]. This tunability makes them incredibly versatile and valuable across a wide range of applications, including catalysis, sensing, imaging, drug delivery, electronics, and environmental remediation [10,11]. The synthesis of MBNPs can be achieved through various methods, such as chemical reduction, sol–gel processes, electrochemical deposition, and laser ablation, among others. Each method offers unique advantages in terms of particle size distribution, shape control, and scalability. Due to their small size and large surface area, MBNPs possess enhanced catalytic activity, making them indispensable in catalysis for chemical transformations, such as hydrogenation, oxidation, and carbon-carbon bond formation [12,13]. Additionally, their unique optical properties, such as localized surface plasmon resonance (LSPR), enable applications in sensing, imaging, and photothermal therapy (PTT) [14,15,16].

In the relentless battle against cancer, the request for more effective and targeted treatment approaches has led researchers to explore the remarkable potential of nanotechnology [17]. Among the myriads of nanoparticles under investigation, MBNPs have emerged as particularly promising candidates in the field of cancer therapy [18]. Defined by their unique physical and chemical properties at the nanoscale, these particles offer a diverse array of functionalities that hold great promise for revolutionizing cancer treatment. With their ability to be precisely engineered and tailored for specific applications, MBNPs present a good opportunity to address the inherent challenges of conventional cancer therapies, such as off-target effects and drug resistance.

MBNPs offer versatile and promising strategies for cancer treatment across multiple fronts, including radiation enhancement, diagnostic imaging, and immunomodulation [19]. Gold and iron oxide nanoparticles, for example, are extensively used as contrast agents for imaging modalities like magnetic resonance imaging (MRI), computed tomography (CT), and surface-enhanced Raman scattering (SERS). These nanoparticles enhance the sensitivity and specificity of imaging techniques, enabling early detection of tumors and accurate assessment of their size, location, and progression [20]. MBNPs can be useful as carriers for chemotherapeutic drugs, peptides, or nucleic acids, facilitating targeted drug delivery to cancer cells. By functionalizing the surface of nanoparticles with targeting ligands, such as antibodies or aptamers, specific recognition and uptake by cancer cells are achieved, minimizing systemic toxicity and enhancing therapeutic efficacy [21].

In PTT, gold nanoparticles, due to their unique optical properties and high photothermal conversion efficiency, are widely employed. When irradiated with near-infrared (NIR) light, gold nanoparticles selectively accumulate in tumor tissue and convert light energy into heat, leading to localized hyperthermia and the thermal ablation of cancer cells while sparing surrounding healthy tissue [22,23,24].

Other types of MBNPs, including gold and platinum nanoparticles, act as radiosensitizers to enhance the efficacy of radiation therapy. When administered prior to radiation treatment, these nanoparticles increase the sensitivity of cancer cells to ionizing radiation, resulting in enhanced DNA damage and cell death within the tumor microenvironment [25,26].

Furthermore, MBNPs possess immunomodulatory properties that can be exploited to enhance the effectiveness of immunotherapy approaches in cancer treatment. By modulating the tumor microenvironment and promoting antitumor immune responses, these nanoparticles can increase the efficacy of immune checkpoint inhibitors, cancer vaccines, and adoptive cell therapies, leading to improved therapeutic outcomes [27]. Additionally, they enable theranostic applications, integrating diagnostic and therapeutic functionalities into a single platform. By combining imaging agents with therapeutic cargo, such as drugs or nucleic acids, these nanoparticles allow for the real-time monitoring of the treatment response while simultaneously delivering targeted therapy to cancer cells, thus facilitating personalized medicine approaches in oncology (better described in the last paragraph) [28,29].

## 3. PDT: Mechanism of Action and Applications

PDT is a minimally invasive treatment modality primarily used in oncology to selectively target and destroy cancer cells [30]. Its mechanism of action involves the interaction of three key components: a photosensitizer (PS), light of a specific wavelength, and molecular oxygen. The process begins with the administration of a photosensitizing agent, typically a dye or a light-sensitive drug, either orally, intravenously, or topically. The PS preferentially accumulates in target tissues, including cancer cells, due to their increased vascularity and altered metabolism. Once administered, the PS accumulates within the target tissue over a period, with higher concentrations found in cancer cells compared to the surrounding healthy tissues. This selective uptake is attributed to various factors, including the enhanced permeability and retention (EPR) effect and specific cellular uptake mechanisms. In particular, EPR is a phenomenon where nanoparticles and macromolecules preferentially accumulate in tumor tissue due to the tumor’s leaky vasculature and poor lymphatic drainage. This allows for targeted drug delivery in cancer therapy [31]. After a sufficient accumulation period, the target tissue is exposed to light of a specific wavelength that corresponds to the absorption spectrum of the PS. This light can be delivered externally using lasers or light-emitting diodes (LEDs) and is typically in the visible or near-infrared (NIR) range. Upon exposure to light, the PS in an excited state interacts with molecular oxygen (O_2_) present in the tissue, leading to the production of reactive oxygen species (ROS), primarly singlet oxygen (^1^O_2_) and free radicals [32,33]. The details of the PDT mechanism are better explained in Figure 1.

This process, known as photoactivation, is the key cytotoxic mechanism of PDT. The generated ROS, particularly singlet oxygen, causes oxidative damage to cellular components such as lipids, proteins, and DNA within the target cells [34]. This oxidative stress triggers various cellular responses, including programmed and unprogrammed cell death, leading to the destruction of cancer cells [35]. In addition to direct cytotoxic effects, PDT induces immunomodulatory responses within the tumor microenvironment. The localized release of damage-associated molecular patterns (DAMPs) and pro-inflammatory cytokines activates innate and adaptive immune responses, leading to tumor infiltration by immune cells and subsequent tumor regression. Following treatment, the damaged cancer cells (apoptotic or necrotic) are cleared by phagocytic cells of the immune system, such as macrophages [36,37,38]. This clearance process helps in the resolution of inflammation and tissue repair within the treated area. PDT has gained significance in cancer therapy due to its unique approach and advantages over traditional treatments. One key aspect that underscores its importance is its ability to target cancer cells with precision, minimizing damage to surrounding healthy tissues. This precision is achieved through the selective accumulation of PSs in tumor cells, followed by activation with a specific wavelength of light [39]. This mechanism reduces the risk of collateral damage, making PDT a valuable option in areas where preserving healthy tissue is crucial, such as in head and neck cancers or certain gynecological cancers [40,41]. PDT can induce various mechanisms of cell damage, including apoptosis, necrosis, and autophagy [42] (Figure 2). The induction of apoptosis necessitates the preservation of plasma membrane integrity and adequate adenosine triphosphate (ATP) levels. During apoptosis, chromatin compacts, resulting in the formation of apoptotic bodies, while DNA fragmentation takes place as these cells undergo programmed cell death. This process of programmed cellular autolysis is stringently regulated at the level of specific regulatory proteins and corresponding effector enzymes. Proteolytic caspases are pivotal regulators in the apoptotic pathway. The activation of caspases can be initiated through both the extrinsic and intrinsic pathways. In the extrinsic pathway, the apoptotic cascade is triggered by the activation of membrane-bound death receptors, which detect external signals (such as Fas, TNF, DR-4, and DR-5). Conversely, in the intrinsic pathway, often implicated in PDT, apoptotic signals arise from intracellular organelles like mitochondria and lysosomes [43,44]. Necrosis is a passive process that does not necessitate energy expenditure. During necrosis, the plasma membrane’s integrity and permeability are compromised, leading to protein denaturation and the release of intracellular contents into the extracellular environment. Autophagy, on the other hand, involves the formation of membrane-bound vesicles within the cytoplasm, which contains cellular organelles or their fragments. When autophagosomes fuse with lysosomes, the resultant autolysosomes degrade and recycle cellular components [45,46]. Understanding the mechanism of PDT is crucial for designing potential PSs and efficient treatment protocols. PDT kills tumor cells directly through apoptotic and non-apoptotic pathways (necrosis, autophagy) and indirectly by damaging tumor vasculature, which supplies nutrients and oxygen to the tumor cells. PDT can also trigger an immune response through the described mechanisms, offering long-term protection against cancer. These mechanisms are activated by various signaling pathways depending on the type of PS and protocol used, dosage, and PS localization, as well as the genotype of the cells subjected to PDT, oxygen level, and other factors. PSs localized in the mitochondria are likely to induce apoptosis, whereas those localized in the plasma membrane and lysosomes cause necrosis [47].

Furthermore, PDT is important in cancer therapy because of its low systemic toxicity [48]. Unlike chemotherapy or radiation, which often affect the whole body, the effects of PDT are localized. This characteristic means that patients show fewer common side effects like weakness, nausea, or hair loss. This makes PDT a suitable alternative for patients who may not tolerate conventional therapies or those with additional health conditions that limit treatment options. Moreover, PDT is versatile, with applications across a variety of cancer types, including skin, lung, and esophageal cancers, among others [49,50]. This versatility extends to its use in recurrent cancers and as an adjunct to other treatments, providing an additional layer of therapeutic strategy. For example, PDT can be used to treat superficial tumors or as a palliative treatment to relieve symptoms in cases where surgery is not an option. Overall, the importance of PDT in cancer therapy lies in its combination of effectiveness and minimal invasiveness. It offers a targeted approach with reduced systemic side effects, is adaptable to different types of cancer, and can be repeated if necessary. These factors make it an increasingly valuable tool in the oncologic field, contributing to improved outcomes and quality of life for cancer patients [48].

## 4. MBNPs as PSs in PDT

MBNPs have emerged as innovative PSs in PDT, providing new opportunities for enhancing the efficacy and specificity of cancer treatment [51]. Unlike traditional PSs, which are often organic compounds, MBNPs offer unique properties that make them particularly useful in PDT applications. One of the key advantages of MBNPs is their ability to act as platforms for the controlled delivery of PSs. These nanoparticles can be engineered to encapsulate or attach photosensitizing agents, allowing for targeted delivery to specific tissues or tumor sites [52,53]. This targeted approach can increase the accumulation of PSs in cancer cells, leading to more effective treatment outcomes with minimal damage to surrounding healthy tissue. Another important aspect of MBNPs is their enhanced photophysical properties. Metals like gold, silver, and titanium dioxide have strong plasmonic or photocatalytic characteristics, allowing them to effectively absorb and scatter light at specific wavelengths ([48]. This characteristic can improve the activation of PSs and, consequently, the generation of reactive oxygen species (ROS), which are crucial for inducing cell death in PDT (Figure 3).

In addition, MBNPs can be designed to have unique surface properties, allowing for improved stability, biocompatibility, and functionalization with targeting molecules, offering the potential for multimodal imaging and therapy, known as theranostics. Since these nanoparticles can be designed to incorporate imaging agents, they can simultaneously be used for diagnostic purposes and as therapeutic agents in PDT. This dual functionality allows clinicians to visualize the distribution of nanoparticles in real time, ensuring the accurate targeting of the PS and enabling the monitoring of treatment progression. Furthermore, the versatility of MBNPs allows for a range of applications in PDT. They can be tailored to respond to different types of light, including visible, ultraviolet, and near-infrared light, providing adaptability in treatment protocols [54,55,56]. This versatility is crucial for treating a variety of cancer types and tumor locations, where different wavelengths may be required for optimal penetration and activation.

Several studies have been reported on the investigation of MBNPs as PSs in PDT in different formulations. Some of these are made of pure metal NPs, including, but not limited to, silver, copper, gold, iron, magnesium, titanium, platinum, and zinc. Other classes of NPs comprise metal/transition-metal oxide NPs (e.g., MnO_2_, TiO_2_, ZnO, MoO_3_, and CeO_2_), metal sulfide NPs (e.g., Ag_2_S, CuS, and FeS), metal-based nanoformulations doped with other metals (e.g., Au-TiO_2_ or Pt-ZnO) and/or non-metals such as nitrogen and graphene oxide, and metal-organic frameworks (MOFs) [57,58]. Some examples of NPs employed in PDT are described below.

### 4.1. Gold Nanoparticles (AuNPs)

Gold is among the most used materials for nanoparticles in medical applications. They are emerging as important tools to enhance the effectiveness of PDT, thanks to their unique properties and ability to boost various aspects of the treatment [59]. The synthesis of AuNPs involves various methods (Figure 4), each designed to control the size, shape, and surface properties of the nanoparticles. The most common approach is the chemical reduction method, where gold salts, typically gold chloride (HAuCl_4_), are dissolved in water and then reduced using agents such as sodium citrate, ascorbic acid, or sodium borohydride. This reduction process transforms Au^3^⁺ ions into Au^0^, leading to the nucleation and growth of AuNPs, with the reducing agent often serving as a stabilizer to prevent aggregation [60]. Physical methods like laser ablation involve using a laser to ablate a gold target submerged in a liquid medium, while photochemical synthesis relies on light exposure to reduce gold salts in the presence of photo-reducing agents. Biological methods employ extracts from plants, bacteria, fungi, or enzymes, which act as natural reducing and stabilizing agents to produce gold nanoparticles [59,60,61]. For example, Lv et al. (2017) [62] prepared gold nanotriangles (GNTs) using the Yam bean as a green synthesis method and demonstrated that singlet oxygen (^1^O_2_) can be generated when the GNTs are irradiated with 808 nm wavelength light at a low power density of 500 mW cm^−2^. In this work, they investigated the effects of PDT and PTT on the viability of HeLa cells incubated with GNTs. GNTs destructed cancer cells via the synergistic effects of PDT and PTT [62].

AuNPs exhibit exceptional light absorption and scattering due to their unique interaction with light. This strong interaction arises from the collective oscillation of conduction electrons on the metal surface when excited by light at specific wavelengths, known as surface plasmon resonance (SPR) [16]. This phenomenon significantly enhances the absorption and scattering intensities of AuNPs compared to non-plasmonic nanoparticles of the same size. The absorption and scattering properties of AuNPs can be adjusted by manipulating the particle size, shape, and the local refractive index near their surface [63]. These properties increase the energy transferred to the PS, enhancing the generation of reactive oxygen species (ROS) during PDT. Once inside the cell, AuNPs can indeed promote ROS production, boosting the effectiveness of PDT. This increase in ROS can ultimately result in programmed cell death or apoptosis in cancer cells. Furthermore, AuNPs can act as carriers for PSs, allowing for more precise delivery to cancer cells [64]. This increases the concentration of the PS in the target area and reduces its impact on surrounding healthy tissue. Thanks to their optical properties, AuNPs can also be used for therapy monitoring and imaging, providing useful information on the progress of treatment and the effectiveness of PDT [65,66]. For example, they can be used in combination with other therapies, such as PTT, where the heat generated by the nanoparticles upon exposure to light can be used to destroy cancer cells. This synergistic approach can increase the overall treatment efficacy [67,68]. AuNPs offer a versatile platform for surface modification, accommodating a range of agents such as peptides, antibodies, aptamers, and biocompatible polymers like polyethylene glycol (PEG). These modifications enable tailored functionality and enhanced properties, empowering diverse applications in biomedicine, diagnostics, and nanotechnology [69,70,71]. In Wang 2024, AuNPs were modified with a surface alteration using polyethylene glycol (PEG) and lipid-conjugated peptides (pepducin) to enhance stability and specificity toward cancer cells. The research evaluated the stability of three distinct AuNP formulations in different solutions, revealing that PEGylation notably improved nanoparticle stability, while the integration of pepducin enhanced their internalization and affinity for cancer cells [72].

### 4.2. Silver Nanoparticles (AgNPs)

Novel therapeutic strategies for cancer through nanotechnology propose metallic AgNPs as promising candidates for cancer therapy. They have been shown to possess anticancer properties, including the selective inhibition of the mitochondrial respiratory chain, leading to the generation of reactive oxygen species (ROS) and DNA damage [73]. AgNPs are synthesized through the conversion of silver ions using nanotechnology into ultra-small materials measured in nanometers (Figure 5) [74].

The conversion of silver ions to AgNPs through eco-friendly or biogenic methods offers advantages over chemical and physical approaches. For example, the use of plants in biogenic AgNPs synthesis is recognized as highly cost-effective, less hazardous to human health and the environment, and facile to carry out [76]. Experimental evidence suggests that AgNPs can be used in cancer PDT as standalone material-based molecules (PS) or in combination with other PSs. For example, the results from an experimental study utilizing AgNPs for mediating PDT showed that irradiating AgNPs at 635 nm decreased cell proliferation and viability and induced apoptosis in both MCF7 and A549 cancer cells. However, AgNPs exhibited a significantly lower cytotoxic effect on A549 cells compared to MCF7 cells. This indicates that different types of cancer cells can respond variably to the same forms of metallic AgNPs [77]. In another study, the photodynamic therapy activity of a novel metal-free phthalocyanine, both with and without silver nanoparticles (AgNPs), was reported. The in vitro photodynamic activity of this complex was evaluated using a metastatic melanoma cancer cell line. The study demonstrated that the viability of the cells decreased in a concentration-dependent manner when exposed to different concentrations of PSs, either alone or in combination with AgNPs, upon irradiation with red light. The presence of AgNPs enhanced the photodynamic activity of the PS [78].

### 4.3. Titanium Dioxide Nanoparticles (TiO_2_NPs)

Titanium dioxide nanoparticles (TiO_2_NPs) are extensively used in cancer therapy, especially in PDT [79,80,81], due to their remarkable photocatalytic properties. The synthesis of TiO_2_NPs can be accomplished through several methods, each offering different advantages and yielding nanoparticles with distinct properties. The sol–gel method involves dissolving titanium alkoxide in an alcohol solvent, followed by hydrolysis with water, often catalyzed by an acid or base. The resulting sol transforms into a gel, which is then dried and calcined to form TiO_2_NPs. The hydrothermal method mixes a titanium precursor with water in a sealed autoclave, heating the mixture to facilitate nanoparticle formation. Reaction conditions such as temperature and pH are controlled to influence the properties of the nanoparticles. The precipitation method reacts a titanium salt with a basic solution to form a titanium hydroxide precipitate, which is then filtered, dried, and calcined to produce TiO_2_NPs. The microemulsion method creates a microemulsion with water, oil, and a surfactant, where the titanium precursor undergoes hydrolysis within nanometer-sized water droplets, resulting in precise control over nanoparticle size and shape. Each method allows the tailoring of the nanoparticles’ properties for specific applications [82].

When exposed to ultraviolet (UV) light, TiO_2_NPs generate reactive oxygen species (ROS), such as singlet oxygen and hydroxyl radicals. This limitation restricts NP absorption to the UV but not the NIR light region [83]. Despite their advantages, UV irradiation is often unsuitable for PDT due to its limited penetration depth, lower light content, and significant harmful side effects for patients exposed to UV light. Consequently, extensive research has focused on extending the photoresponse of TiO_2_NPs into the visible light spectrum. The surface modification of TiO_2_NPs aims to incorporate biologically active species to enhance the selectivity and therapeutic effectiveness of these nanoparticles. Moreover, such surface modifications have been shown in recent studies to mitigate the potential toxicity associated with unmodified TiO_2_NPs, as better described below.

In PDT, TiO_2_NPsare utilized as PSs [81]. Upon administration, these nanoparticles accumulate in the tumor tissue, benefiting from the EPR effect commonly observed in solid tumors. Once localized in the tumor, the area is irradiated with UV light, activating the TiO_2_NPs. The resultant ROS generation leads to oxidative stress within the cancer cells, causing cellular damage and, ultimately, cell death. Additionally, the high biocompatibility and chemical stability of TiO_2_NPs make them suitable for medical applications. They do not readily degrade or lose their functionality in the biological environment, ensuring a sustained therapeutic effect [84]. Furthermore, their surface can be modified to enhance targeting and improve the selectivity of the treatment, thereby minimizing damage to surrounding healthy tissues [85]. The surface modification not only prevents agglomeration but also allows for further functionalization/conjugation. There are two approaches to achieve this: non-covalent and covalent conjugation of organic (or inorganic) species with TiO_2_NPs. The first strategy relies on physical interactions (such as electrostatic, hydrogen bond, van der Waals, and hydrophobic interactions), offering simplicity and maintaining the structure of the modifiers. Nevertheless, various external factors like temperature, pH, and ionic strength could negatively interact with this form of modification. Covalent conjugation involves bonding modifiers to the surface of TiO_2_NPs by using various agents such as polymers, organophosphorus molecules, carboxylic acids, and organosilanes [86]. Extensive research in the PDT field has been focused on expanding the photoresponse of TiO_2_NPs to the visible light spectrum, as UV light absorption in TiO_2_NPs is very important in triggering redox reactions that generate cancer-fighting reactive oxygen species, but UV irradiation is not ideal for PDT due to potential harm to patients. Therefore, efforts have been made to enhance the photoresponse of TiO_2_NPs to visible light; in this context, there is a growing interest in modifying the surface of TiO_2_ with porphyrins due to the possibility of extending the absorption spectrum of TiO_2_NPs from UV to visible wavelengths. The PS can be conjugated to the TiO_2_NP surface through either non-covalent or covalent methods; this latter method can ensure the stability of the nanosystem, particularly when using silane linkers that have a strong affinity for the hydroxyl groups of TiO_2_NPs. For example, it reported two alternative approaches for attaching Chlorin 6 (Ce6), a potent porphyrin-based PS: TiO_2_NPs were either modified by the growth of a polysiloxane layer made up of two silane reagents ((3-aminopropyl)triethoxysilane (APTES) and tetraethyl orthosilicate (TEOS)) around the core (resulting in PEGylated nanoparticles: TiO_2_@4Si-Ce6-PEG, SiO_2_@4Si-Ce6-PEG) or simply modified by APTES alone (resulting in APTES-modified nanoparticles: TiO_2_-APTES-Ce6, SiO_2_-APTES-Ce6). Ce6 was covalently attached to the modified TiO_2_NPs via an amide bond. In vitro experiments on glioblastoma U87 cells demonstrated the photodynamic efficiency of the new hybrid nanoparticles. In particular, the APTES-modified nanosystems were more efficient than PEGylated TiO_2_NPs and free Ce6 [87]. An alternative way to improve TiO_2_NPs ability to target specific types of cancer cells, enhancing the affinity for ill cells and increasing the effectiveness of PDT treatment is the addition of folic acid (FA), which ensures better penetration of the cell membrane through folate receptors, which are commonly found in cancer cells rather than healthy ones. A study introduced a novel photosensitizer, FA-TiO_2_-Pc, composed of TiO_2_NPs conjugated with FA and Al (III) phthalocyanine chloride tetrasulfonic acid, a promising PS with limitations in tumor targeting and affinity to cancer cells. The complex increased drug delivery both in vitro and in vivo and demonstrated high therapeutic activity and biocompatibility. In vivo testing on mice with HeLa xenograft tumors showed suppressed tumor growth with minimal side effects using a low dose of FA-TiO_2_-Pc and low light exposure [88]. Pan and coworkers modified TiO_2_NPs with semiconductor quantum dots (QDs) CdX (X = S, Te, Se) using various techniques, including ultrasonic, hydrothermal, sol–gel, aqueous phase, and hydrolysis precipitation. This modification expands the absorption range of TiO_2_NPs into different visible light regions successfully. Among the CdX QDs, the CdSe-TiO_2_ nanocomposite activated by visible light demonstrated the highest PDT efficiency against HL60 cells; furthermore, when combined with FA, the complex exhibited excellent cancer-targeting abilities during PDT treatment [89]. Salama developed a complex of TiO_2_NPs ligated through polyethylene glycol to EGF (epidermal growth factor), increasing the selectivity of NPs to cancer cells that expressed EGFR (epidermal growth factor receptor) and increasing PDT efficiency on A431 epidermal cancer cells [90]. Overall, the application of TiO_2_NPs in PDT provides a promising approach for effectively targeting and eliminating cancer cells, leveraging the nanoparticles’ unique properties to enhance the therapeutic outcomes of the treatment.

### 4.4. Magnetic Nanoparticles (MNPs)

MNPs play a crucial role in enhancing PDT due to their unique magnetic and surface properties. The main types of MNPs used in PDT include iron oxide nanoparticles (such as Fe_3_O_4_ and γ-Fe_2_O_3_), which are widely utilized because of their biocompatibility and strong magnetic properties [91]. These nanoparticles, particularly in the form of superparamagnetic iron oxide nanoparticles (SPIONs), can be easily manipulated by external magnetic fields, making them ideal for targeted drug delivery and imaging [92]. Other types of MNPs include metallic nanoparticles like cobalt (Co), nickel (Ni), and iron (Fe), which offer higher magnetic moments and can enhance magnetic targeting, although their biocompatibility and potential toxicity must be carefully managed. Additionally, ferrites such as MnFe_2_O_4_, ZnFe_2_O_4_, and CoFe_2_O_4_ can be used; their magnetic properties can be tuned by altering the metal ion, offering good biocompatibility and magnetic responsiveness. Key properties of MNPs that make them suitable for PDT include their superparamagnetism, which ensures strong magnetic behavior only in the presence of an external magnetic field and minimizes particle aggregation when the field is removed. This property also enables these nanoparticles to serve as contrast agents in magnetic resonance imaging (MRI), aiding in the diagnosis and monitoring of treatment progress [93]. The surface functionalization of MNPs with biocompatible materials like silica, polymers, or lipids improves their stability, reduces toxicity, and allows for the conjugation of PSs, targeting ligands, or therapeutic agents [94]. Functionalization with targeting ligands such as antibodies or peptides facilitates selective targeting of cancer cells, thereby enhancing the specificity and efficiency of PDT. Figure 6 shows an example of functionalized MNPs under a transmission electron microscope, in particular, a conjugate between amino-functionalized silica magnetite and the siderophore feroxamine used for the biological field (A) and another type of functionalization of MNPs for different biological applications (B).

In addition to these properties, some MNPs exhibit photothermal effects, where they convert absorbed light into heat, providing a dual-mode therapy that combines PDT and PTT [97]. This combination can lead to more effective tumor destruction by leveraging both light-induced cytotoxic effects and localized heating. Overall, the integration of MNPs into PDT offers significant advantages in terms of targeting, imaging, and therapeutic efficacy [91].

## 5. Challenges and Limitations in MBNP-Mediated PDT

MBNPs have significant potential to enhance PDT, but several challenges and limitations hinder their clinical application. One major issue is toxicity and biocompatibility. Certain metal nanoparticles, like those made from cobalt or nickel, can induce cytotoxic effects due to the leaching of metal ions, which interfere with cellular processes [98]. Even nanoparticles generally considered biocompatible, such as iron oxide, require thorough evaluation to ensure they do not cause harm over prolonged exposure [99]. The stability and aggregation of MBNPs also pose significant challenges. These nanoparticles can aggregate in biological environments, which affects their stability and alters their distribution within the body [100,101]. This aggregation can hinder the effective delivery and activation of the PS in the tumor, reducing the overall efficacy of PDT. Surface modification is a crucial strategy to overcome this limitation. MBNPs tend to aggregate due to their high surface energy, which can lead to the loss of their unique properties and reduced effectiveness in various applications. Several methods are used for surface modification to prevent aggregation. Steric stabilization involves coating nanoparticles with polymers such as polyethylene glycol (PEG), polyvinylpyrrolidone (PVP), or block copolymers, providing a steric barrier that prevents close contact between nanoparticles [102,103]. Similarly, attaching organic ligands with long hydrophobic tails to the nanoparticle surface creates a steric hindrance that keeps the particles separated. Electrostatic stabilization can be achieved by adsorbing charged molecules like citrate ions or surfactants onto the surface of nanoparticles, imparting a surface charge that creates electrostatic repulsion between particles. For instance, in the case of gold nanoparticles, a common practice is to use citrate as a capping agent. The citrate ions impart a negative charge to the surface, providing electrostatic repulsion that prevents aggregation [104,105]. Adjusting the pH of the nanoparticle suspension to ensure that the nanoparticles carry a uniform charge can also enhance electrostatic repulsion [106,107,108].

Encapsulation in a polymer matrix, such as polydopamine or silica, provides a robust barrier that prevents nanoparticles from coming into direct contact. Moreover, cross-linking polymer chains on the nanoparticle surface create a network that prevents the particles from coming too close together. Using multifunctional ligands that can form cross-links between nanoparticles helps maintain a stable dispersion [109]. Choosing appropriate solvents like ethanol, ethylene glycol, or aqueous buffers that can stabilize nanoparticles through solvation forces also helps prevent aggregation.

By employing these surface modification techniques, the stability and functionality of metal nanoparticles can be significantly improved, making them more effective for various applications such as drug delivery, catalysis, sensing or enhancing of PDT or PTT. [110,111]. Some examples of outcomes using surface-modified MNPs are reported in Table 1.

In addition to improving stability and biocompatibility, the surface modifications of MBNPs can improve targeting by modifying their surface properties to enhance specificity and efficiency in reaching and interacting with target cells, typically cancer cells. This can be achieved through active targeting, which involves attaching specific ligands to the surface of the NPs. These ligands can bind to receptors overexpressed on the surface of target cells, ensuring that the nanoparticles accumulate primarily in the desired location. Common ligands include (i) antibodies, which are proteins designed to bind specifically to antigens present in cancer cells, significantly improving targeting efficiency [120]; (ii) peptides, which are short chains of amino acids used as targeting ligands due to their ability to recognize and bind to specific cellular receptors [121]; (iii) small molecules such as folic acid targeting folate receptors, which are often overexpressed on cancer cells [122]; (iv) oligonucleotides, short DNA or RNA sequences, designed to target specific genetic sequences within cancer cells [123]; (v) vitamins, which are essential nutrients, used as targeting agents due to their specific uptake mechanisms in cells [124]; (vi) signal peptides, which direct the transport of proteins within cells and are utilized for their ability to guide NPs to specific cellular compartments [125]. Another approach is passive targeting, which relies on the enhanced permeability and retention (EPR) effect. This effect exploits the leaky vasculature and poor lymphatic drainage of tumor tissues, allowing nanoparticles to accumulate in tumors more effectively. However, this method can be less precise compared to active targeting.

Functionalizing nanoparticles with targeting ligands, such as antibodies or peptides, aims to enhance specificity for tumor tissues. However, the heterogeneous nature of tumors and their complex microenvironments can impede the uniform distribution and penetration of these nanoparticles. Additionally, the use of targeting ligands may provoke immune responses or cause off-target effects, complicating their safe and effective use [126].

Combining MBNPs in PDT with other strategies can create a synergistic therapeutic approach to cancer treatment. Numerous studies have explored combining PDT with photothermal therapy [127], sonodynamic therapy [128], chemodynamic therapy [127], chemotherapy [129], starvation therapy [130], radiotherapy [131], immunotherapy [132], and gas therapy [133]. By integrating PDT with these diverse cancer treatment modalities, researchers aim to leverage their respective strengths while mitigating their weaknesses, thereby achieving synergistic benefits (Figure 7).

MBNPs, such as gold nanoparticles, are excellent candidates for both PDT and PTT due to their ability to absorb and convert light energy into heat (PTT) and generate reactive oxygen species (ROS) upon activation by light (PDT). In this combined approach, metal nanoparticles can be designed to absorb light at specific wavelengths suitable for both PDT and PTT [66,135,136]. When exposed to light, these nanoparticles generate heat, which can enhance the cytotoxic effects of PDT by further damaging cancer cells through hyperthermia. This dual treatment modality not only increases tumor cell killing but also offers the potential to overcome resistance mechanisms that may limit the efficacy of each therapy alone. Balancing the photothermal and photodynamic effects of MBNPs is crucial yet challenging. These nanoparticles can convert absorbed light into heat (photothermal effect) while also generating ROS for PDT. While this dual functionality can enhance therapeutic outcomes, it requires precise control to ensure that the thermal effects do not damage surrounding healthy tissues or reduce the efficacy of the ROS generated for PDT. Regulatory and manufacturing challenges also limit the widespread adoption of MBNPs in clinical settings. The complex synthesis and functionalization processes must meet stringent regulatory standards to ensure safety, efficacy, and reproducibility. Scaling up these processes for large-scale production without compromising quality is another significant hurdle.

## 6. Conclusions: Future Directions and Opportunities in Research for Cancer Therapy

In summary, while MBNPs hold promise for improving PDT, their clinical application is limited by challenges related to toxicity, stability, targeting efficiency, and the balance of therapeutic effects, as well as regulatory and manufacturing issues. Furthermore, the integration of MBNPs in PDT for cancer treatment holds great potential to revolutionize oncological therapies. Advances in nanoparticle engineering have enabled enhanced targeting, improved delivery of PSs, and reduced off-target effects. The synergistic combination of PDT with other therapeutic modalities, such as photothermal therapy, chemotherapy, and immunotherapy, can enhance treatment efficacy and overcome the limitations of conventional therapies. The development of stimuli-responsive nanoparticles and the incorporation of imaging techniques provide promising avenues for precise, real-time monitoring and tailored treatment strategies. Addressing these challenges requires continued research and development to optimize the design and use of these nanoparticles in medical applications.

## Figures and Tables

**Figure 1 pharmaceutics-16-00932-f001:**
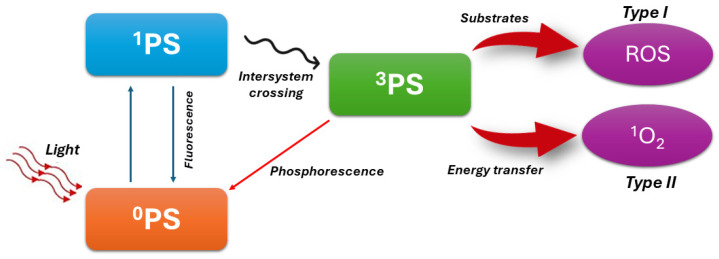
Basic PDT type I and type II process schematic diagram. In the type I process, the activated PS interacts directly with cellular components such as lipids, proteins, and nucleic acids. This interaction leads to the transfer of electrons or hydrogen atoms from these cellular components to the excited PS. The transfer results in the formation of ROS, including superoxide anion radicals (O_2_^−^•), hydroxyl radicals (•OH), and other radical species. In the type II process, the excited PS transfers its energy directly to molecular oxygen (^3^O_2_) in the surrounding tissue. This energy transfer converts the ground state oxygen (triplet state, ^3^O_2_) to an excited state known as singlet oxygen (¹O_2_). ^0^PS: PS ground single state; ^1^PS: PS excited singlet state; ^3^PS: PS excited triplet state; ROS: reactive oxygen species; ^1^O_2_: singlet oxygen.

**Figure 2 pharmaceutics-16-00932-f002:**
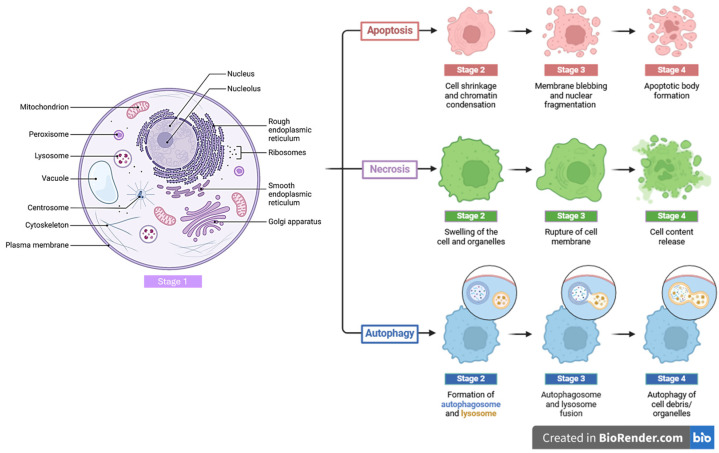
Structural overview of an animal cancer cell with its components contributing to its functionality (**left**). Major pathways of cell death through apoptosis, necrosis, and autophagy, which describe the main processes involved in each pathway (**right**). This figure was created using BioRender.com.

**Figure 3 pharmaceutics-16-00932-f003:**
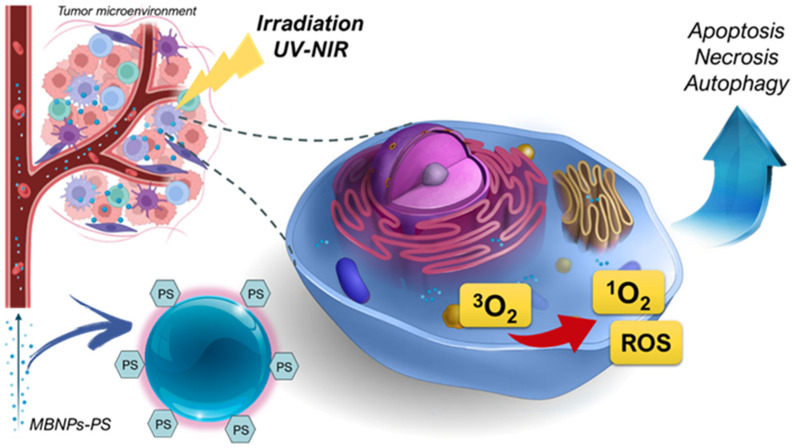
Biodistribution of MBNPs linked to PS within the tumor microenvironment and the generation of reactive oxygen species (ROS) and singlet oxygen (^1^O_2_) upon irradiating the tumor site with an appropriate wavelength of light. During PDT, NPs enhance oxygen production, leading to a cascade of events that induce programmed cell death in tumor cells.

**Figure 4 pharmaceutics-16-00932-f004:**
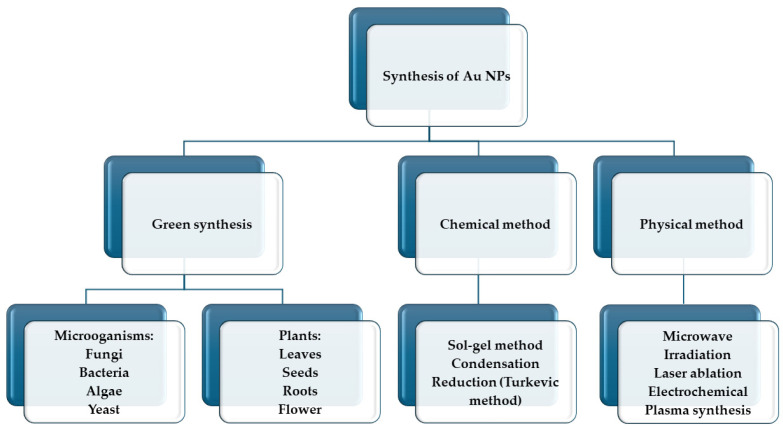
Different methods for the synthesis of AuNPs.

**Figure 5 pharmaceutics-16-00932-f005:**
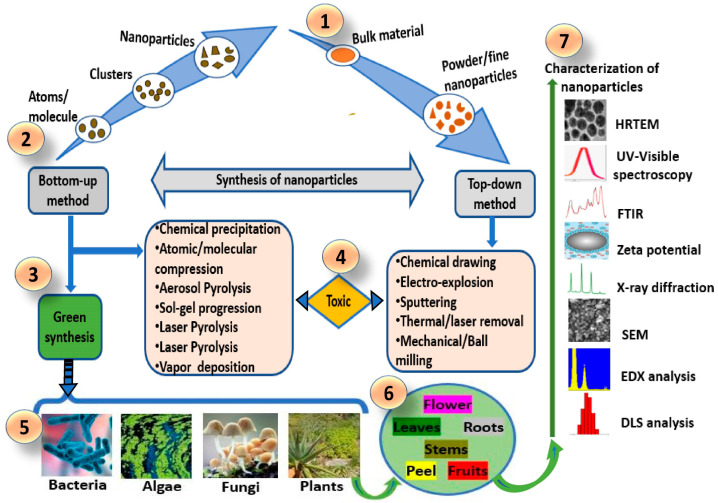
Bottom-up and top-down methods for nanomaterial synthesis. (1) Synthesis using bulk material in the top-down method; (2) synthesis using atomic structures/molecules in the bottom-up method; (3) green synthesis approaches in bottom-up methods; (4) toxic method for nanomaterial synthesis in bottom-up and top-down methods using physical and chemical approaches; (5) biological sources exploited in the bioformulation of biogenic (green) nanomaterials; (6) biological plant parts that are used in biogenic nanomaterial synthesis (7); characterization techniques to confirm the synthesis of nanomaterials. Reproduced with permission from Kah et al., Cells, 2023 [75].

**Figure 6 pharmaceutics-16-00932-f006:**
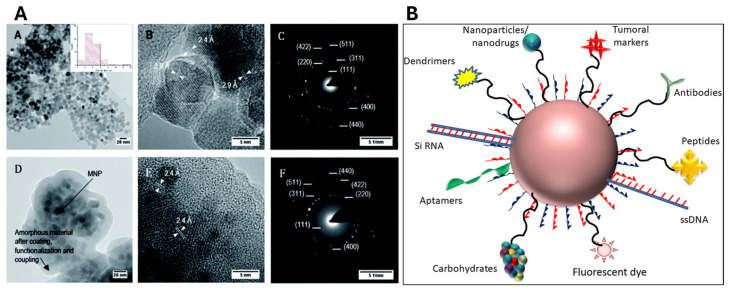
(**A**) Bright field TEM images and electron diffraction of bare MNPs (**A**–**C**) and of MNPs functionalized with feroxamine (**D**–**F**) [95]. Reproduced with permission from the Royal Society of Chemistry. (**B**) Representation of functionalized nanoparticles and different attached targeting molecules for property amelioration [96]. Reproduced with permission from the Royal Society of Chemistry.

**Figure 7 pharmaceutics-16-00932-f007:**
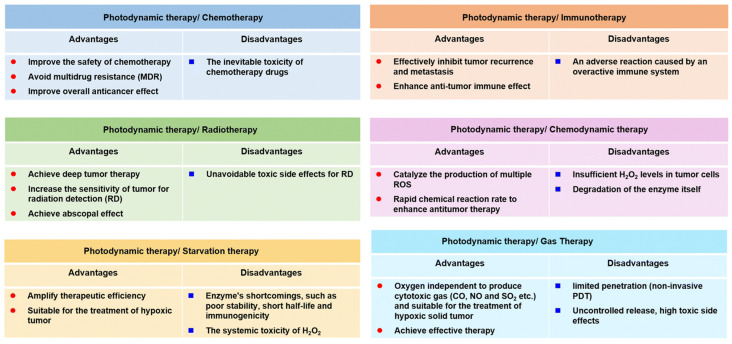
Advantages and disadvantages of synergistic therapies containing PDT [134]. Reproduced with permission from the Royal Society of Chemistry.

**Table 1 pharmaceutics-16-00932-t001:** Effects of surface modifications in different MBNPs used in cancer therapy.

MNPs	Surface Modification	In Vitro System	Highlights	Reference
Gold NPs	Curcumin, isoniazid, tyrosine, and quercitin	Raw 264.7 cells	The peroxidase-mimicking nanoparticle interactions with red blood cells and mouse macrophages confirmed their hemocompatible and biocompatible nature	[112]
Gold NPs	Epidermal growth factor (EGF)	MDA-MB-468 cells and MCF7cells	111In-EGF-Au NPs were significantly more radiotoxic to MDA-MB-468 than MCF-7 cells with a surviving fraction of 17.1 ± 4.4% versus 89.8 ± 1.4% (*p* < 0.001) after exposure for 4 h	[113]
Silver NPs	Glucose, lactose, and oligonucleotides	L929 and A549 cells	The binding of oligonucleotides along with the carbohydrate on the AgNP surfaces influenced the differential uptake rate pattern into the cells. The cytotoxicity study with the modified AgNPs revealed that only naked AgNPs influence the viability of A549 cells	[114]
Palladium NPs	Graphene oxide	PC3 cells	Compared to GO or Pd NPs alone, GO-Pd NPs showed higher cytotoxic effects in prostate cancer 3 (PC3) cells. The irradiation of treated cells with a near-infrared (NIR) laser considerably enhanced apoptosis induced by the synergistic photothermal effect and reactive oxygen species (ROS) generation	[115]
Palladium NPs	Transferrin	MCF7 cells	The combination of phototherapy induced by PdNPs and a chemotherapeutic agent (PTX) could exhibit synergistic anticancer activities.	[116]
Platinum NPs	Bovine serum albumin	4T1 cells	The results showed a greater cytotoxic effect compared to cells treated with only the BSA-PtNPs, suggesting that these nanomaterials may act as a potential radiosensitizer by improving the efficacy of radiotherapy	[117]
Silver NPs	Doxorubicin	MCF7 cells and MDA-MB-231 cells	The effect was mediated by activation of the tumor suppressor gene (PTEN), which restricts the PI3K/AKT signaling pathway, leading to mitochondrial dysfunction and the activation of caspases three and nine, ultimately resulting in cell apoptosis	[118]
Cobalt ferrite NPs	Polyethylene glycol (PEG)	Lymphocytes	The cytotoxicity of PEG-encapsulated MNPs was better than the bare particles and showed very low toxicity values.	[119]

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
