# Peer review of "Recent Advances in Photodynamic Therapy: Metal-Based Nanoparticles as Tools to Improve Cancer Therapy"

_pharmaceutics, 2024, doi:10.3390/pharmaceutics16070932_

Round 1
Reviewer 1 Report
Comments and Suggestions for Authors
In the review entitled “Recent advances in photodynamic therapy: metal-based nanoparticles as tools to improve cancer therapy.” By Stefania Mariano, Elisabetta Carata and Elisa Panzarini, the authors try to provide a comprehensive understanding of the role of metal-based nanoparticles in photodynamic therapy. The authors refer that they explore the mechanisms, biocompatibility, and applications of metal-based nanoparticles in photodynamic therapy, highlighting the challenges and the limitations in their use, as well as the combining of metal-based nanoparticles/photodynamic therapy with other strategies as a synergistic therapeutic approach for cancer treatment.
Although the review is likely to interest a reasonable number of researchers, the topics are not discussed with adequate depth and are too vague. Additionally, the review would be more appealing and didactic if the authors complemented the information with schemes, figures, etc. They can request permission from the editors of the mentioned journals to do so.
Here are some aspects and considerations to help the authors reformulate the article. The order follows the appearance in the manuscript and is not related to their importance.
i) In figure 1 indicate molecular oxygen O2 as 3O2
ii) Also indicate the 0 in PS as superscript (0PS)
iii) Line 155 what the authors want to mean “This process, known as photoactivation or photobleaching, is the key cytotoxic mechanism of PDT# Since photobleaching refers to the degradation or inactivation of photosensitizer molecules and although this sometimes happen (but PDT process will stop) it is expected that the PS is able to recycle several times.
iv) In section 3, I suggest that the authors complement the description of the types of cell damage with some simple schematic representations. This will help readers who are not familiar with the subject and want to learn something new.
v) Line 255 Gold nanoparticles should be in a different line
vi) This section should be complemented with some schemes showing the construction of the gold nanoparticles. The description is too vague and without any concrete examples.
vii) The same comment for silver nanoparticles. Complete this section with some schemes. The authors can ask permission to the editors as mentioned above.
viii) In section of TiO2 nanoparticles how they are prepared?
ix) Line 322, how their surface can be modified to enhance targeting and improve the selectivity of the treatment, thereby minimizing damage to surrounding healthy tissues [76]?
x) Line 343 In the section of MNPS some figures should be introduced to exemplify the following comment “Surface functionalization of MNPs with biocompatible materials like silica, polymers, or lipids improves their stability, reduces toxicity, and allows for the conjugation of PSs, targeting ligands, or therapeutic agents [80]. Functionalization with targeting ligands such as anti-bodies or peptides facilitates selective targeting of cancer cells, thereby enhancing the specificity and efficiency of PDT.
These are only some examples, and I believe based on this the authors will be able to improve the other parts of the review.
So, although the article is well-written, the authors must make an extra effort to make it more appealing in order to captivate the attention of high number of readers, namely those who are not familiar with the topic.
Author Response
In the review entitled “Recent advances in photodynamic therapy: metal-based nanoparticles as tools to improve cancer therapy.” By Stefania Mariano, Elisabetta Carata and Elisa Panzarini, the authors try to provide a comprehensive understanding of the role of metal-based nanoparticles in photodynamic therapy. The authors refer that they explore the mechanisms, biocompatibility, and applications of metal-based nanoparticles in photodynamic therapy, highlighting the challenges and the limitations in their use, as well as the combining of metal-based nanoparticles/photodynamic therapy with other strategies as a synergistic therapeutic approach for cancer treatment.
Although the review is likely to interest a reasonable number of researchers, the topics are not discussed with adequate depth and are too vague. Additionally, the review would be more appealing and didactic if the authors complemented the information with schemes, figures, etc. They can request permission from the editors of the mentioned journals to do so.
Here are some aspects and considerations to help the authors reformulate the article. The order follows the appearance in the manuscript and is not related to their importance.
i) In figure 1 indicate molecular oxygen O2 as 3O2 WE HAVE DONE
ii) Also indicate the 0 in PS as superscript (0PS) WE HAVE DONE
iii) Line 155 what the authors want to mean “This process, known as photoactivation or photobleaching, is the key cytotoxic mechanism of PDT# Since photobleaching refers to the degradation or inactivation of photosensitizer molecules and although this sometimes happen (but PDT process will stop) it is expected that the PS is able to recycle several times.
Thank you for pointing this out. We agree with this comment. “Photobleaching” refers to the irreversible decomposition of fluorescent molecules; in this case we referred to the activation of a photosensitizing agent by light of a specific wavelength. We deleted the term “Photobleaching”. Line 163.
iv) In section 3, I suggest that the authors complement the description of the types of cell damage with some simple schematic representations. This will help readers who are not familiar with the subject and want to learn something new.
We agree with this comment. We have added a schematic figure that explains the components of a cell and the major pathways of cell death, detailing the steps for each pathway. Line 220
v) Line 255 Gold nanoparticles should be in a different line. WE HAVE DONE. Line 297
vi) This section should be complemented with some schemes showing the construction of the gold nanoparticles. The description is too vague and without any concrete examples.
We thank the referee. We agree with this comment. We have added a schematic figure about the different methods of synthesis of gold nanoparticles, also underlined in the text with some examples. Lines 300-321
vii) The same comment for silver nanoparticles. Complete this section with some schemes. The authors can ask permission to the editors as mentioned above.
We agree with this comment. We have improved the paragraph with a figure explaining the different methods of synthesis of silver nanoparticles described in the text. As suggested by the reviewer, we reused an explanatory and complete figure from Kah et al., 2023. Line 370.
viii) In section of TiO2 nanoparticles how they are prepared?
We thank the referee for this point. We improved this section with the explanation of the methods of synthesis of TiO2 nanoparticles and other concepts included in the next comment. Lines 407-499
ix) Line 322, how their surface can be modified to enhance targeting and improve the selectivity of the treatment, thereby minimizing damage to surrounding healthy tissues [76]?
We thank the referee for the question that allowed us to better improve our manuscript. We reported in the text the literature data about the efficacy of surface modification for improving PDT effectiveness. Lines 442-499
x) Line 343 In the section of MNPS some figures should be introduced to exemplify the following comment “Surface functionalization of MNPs with biocompatible materials like silica, polymers, or lipids improves their stability, reduces toxicity, and allows for the conjugation of PSs, targeting ligands, or therapeutic agents [80]. Functionalization with targeting ligands such as anti-bodies or peptides facilitates selective targeting of cancer cells, thereby enhancing the specificity and efficiency of PDT.
We thank the referee for this comment. We added a figure including two images derived from two papers. The first shows an example of functionalized magnetic nanoparticles, visible under a transmission electron microscopy. The second one shows a diagram of the possible functionalization of MNPs. Line 533
These are only some examples, and I believe based on this the authors will be able to improve the other parts of the review.
So, although the article is well-written, the authors must make an extra effort to make it more appealing in order to captivate the attention of high number of readers, namely those who are not familiar with the topic.
We thank the referee for this comment. We implemented the manuscript based on the suggested comments. Other parts of the work have been improved, also based on the comments of the other referees, hoping that it is now well structured and complete.

Reviewer 2 Report
Comments and Suggestions for Authors
The manuscript is a review on using metal-based nanoparticles in photodynamic therapy (PDT) against cancer. This is very topical and relevant subject. The manuscript is well written, easy to read; language and style are fine. The review addresses all the main points and there is a critical analysis. My only concern is its readership because there are already many similar reviews (e.g. a very recent one: Pashootan et al. Int J Pharm. 2024 Jan 5:649:123622, doi: 10.1016/j.ijpharm.2023.123622).
Other minor issues that need to be resolved are:
There is a large section from the line 128 to 144 with only one reference. It would be useful to add more references that are appropriate for this section. Furthermore, in this section and later in the text EPR effect is mentioned. This concept is not fully accepted, so it would be good to comment on it further.
Please correct in the Line 155 – photoactivation is not the same process as photobleaching.
Line 258 – The title of the subsection should be written in the next line.
Line 270 – it would be useful to describe the main optical properties of AuNPs.
Line 312 – the main advantage of PDT is that it uses harmless visible light. Titanium dioxide NPs use UV light which is not harmless. That could be commented on here.
Author Response
The manuscript is a review on using metal-based nanoparticles in photodynamic therapy (PDT) against cancer. This is very topical and relevant subject. The manuscript is well written, easy to read; language and style are fine. The review addresses all the main points and there is a critical analysis. My only concern is its readership because there are already many similar reviews (e.g. a very recent one: Pashootan et al. Int J Pharm. 2024 Jan 5:649:123622, doi: 10.1016/j.ijpharm.2023.123622).
We thank the referee for this comment. We understand the referee's thoughts very well. The review reported in the comment is focused on cell death pathways such as necroptosis, ferroptosis, cuproptosis, pyroptosis, parthanatos, and immunogenic cell death related to PDT. In this work we tried to describe the different types of metal nanoparticles widely used in PDT, reporting the most recent works with reference to MNPs properties, their use in biological field and the possibility of using them in combination with other therapies. We hope it can be useful for those who are not familiar with the topic.
Other minor issues that need to be resolved are:
There is a large section from the line 128 to 144 with only one reference. It would be useful to add more references that are appropriate for this section. Furthermore, in this section and later in the text EPR effect is mentioned. This concept is not fully accepted, so it would be good to comment on it further.
We agree with this comment. We only included one reference as this part of the text was only a description of the PDT process. However, as suggested by the referee, we have added some references, together with an explanation of the concept of EPR. Line 141
Please correct in the Line 155 – photoactivation is not the same process as photobleaching.
Thank you for pointing this out. We agree with this comment. “Photobleaching” refers to the irreversible decomposition of fluorescent molecules; in this case we referred to the activation of a photosensitizing agent by light of a specific wavelength. We deleted the term “Photobleaching”. Line 163
Line 258 – The title of the subsection should be written in the next line. We have done.
Line 270 – it would be useful to describe the main optical properties of AuNPs.
We thank the referee for this comment. We added a part to the text to describe the optical properties of AuNPs. Line 322
Line 312 – the main advantage of PDT is that it uses harmless visible light. Titanium dioxide NPs use UV light which is not harmless. That could be commented on here.
Thank you for pointing this out. We agree with this comment. We explain the possibility to overcome this limitation by modifying the surface of nanoparticles. Lines 425-499.

Round 2
Reviewer 1 Report
Comments and Suggestions for Authors
The authors considered some of my concerns and the inclusion of new figures and considerations improved the overall clarity of the document.
Comments on the Quality of English LanguageThe english is fine with some minor typos easily corrected during the processament of the manuscript